# Judgments of Event Centrality as Predictors of Post-Traumatic Growth and Post-Traumatic Stress after Infidelity: The Moderating Effect of Relationship Form

Bridget N. Jules [1,*], Victoria L. O'Connor [1,2,3,4] and Jennifer Langhinrichsen-Rohling [1]

[1] Department of Psychological Science, University of North Carolina at Charlotte, Charlotte, NC 28223, USA; victoria.oconnor@va.gov (V.L.O.); jlanghin@charlotte.edu (J.L.-R.)
[2] W.G. (Bill) Hefner VA Healthcare System, Salisbury, NC 28144, USA
[3] VA Mid-Atlantic Mental Illness Research, Education, and Clinical Center (MA-MIRECC), Salisbury, NC 28144, USA
[4] Research & Academic Affairs Service Line, Salisbury Veterans Affairs Medical Center, Salisbury, NC 28144, USA
* Correspondence: bjules1@charlotte.edu

**Abstract:** Infidelity, a betrayal within a romantic partnership, often violates a person's core beliefs about themselves and their significant other and can influence the degree to which a person can feel safe in romantic relationships. Infidelity can also increase exposure to sexually transmitted diseases that can compromise physical and mental health. Therefore, infidelity can be judged as central to one's identity and potentially traumatic, possibly triggering outcomes similar to other DSM-5 Criterion A traumas. The current research examines the contribution of centrality perceptions to the development of PTG and PTS post-infidelity. Bivariate regressions examined the relationships between the judged centrality of infidelity and PTG and PTS, respectively. Exploratory analyses considered the moderating role of relationship form (i.e., casually dating, exclusively dating, and engaged/married) on those relationships. In a sample of 177 adults, greater judgments of the centrality of infidelity were associated with both PTG and PTS. Results demonstrated a significant moderating effect of relationship form on the relationship between the centrality of infidelity and PTG but not between the centrality of infidelity and PTS. Moderation results demonstrated that if infidelity is considered central in a casually dating relationship, it is more strongly related to PTG than in other relationship forms. Considering infidelity as central may generate both beneficial and problematic post-traumatic outcomes. However, an early infidelity experience may provide increased opportunities for engagement in different behaviors in the future (e.g., selecting a different partner, setting different relational boundaries), which, in turn, may be more conducive to growth.

**Keywords:** infidelity; event centrality; post-traumatic growth; post-traumatic stress

## 1. Introduction

Two important, while relatively uncommon, outcomes of trauma include experiences of post-traumatic growth (PTG) and symptoms of post-traumatic stress (PTS; [1]). PTG is characterized by increased personal strength, a renewed sense of life purpose and meaning, and better interpersonal relationships [2]. In contrast, PTS is characterized by intrusive and unwanted memories of the traumatic experience, uncomfortable physical reactions including sweating and heart racing in response to trauma reminders or triggers, and more [3]. PTS is also commonly comorbid with heightened experiences of depression and anxiety [3]. While these two outcomes seem contradictory in their presentation, there is growing recognition that PTG and PTS are linked by similar underlying mechanisms and often co-occur in response to the same stressor [4]. Specifically, prior literature has identified several key cognitive variables as necessary prerequisites for the development of both PTG

and PTS [5]. These include judgments of the centrality of the potentially traumatic event to one's life, as well as perceptions that the event functioned to violate a core belief about oneself, others, and/or the world. However, judgments of event centrality have received relatively less attention in the literature compared to the impact of violated assumptions. Thus, the current study was designed to consider how perceptions of event centrality relate to PTG and PTS after experiencing a potentially traumatic event that has been frequently overlooked in the DSM as well as within the PTG and PTS literature: relationship infidelity.

## 1.1. Event Centrality and Its Connection to PTG and PTS

The centrality of an event has been defined as "the degree to which an individual believes a negative event has become a core part of their identity" [6] (p. 107). Theoretically, this cognitive mechanism is an essential part of an individual's progression towards both post-trauma outcomes, as noncentral events are not important enough to generate a strong or sustained post-traumatic response that requires cognitive attention (i.e., leading to either PTG or PTS). Consistent with this logic, greater perceived event centrality after a range of potentially traumatic events has been associated with increased levels of negative mental health consequences, including depression and PTS [7]. Likewise, judgments of event centrality have emerged as a unique theoretical and statistical predictor of PTG and PTS after experiencing an array of well-studied traumatic events, including death, serious accidents, physical or sexual assault, and wartime combat [8].

As PTG and PTS outcomes have been documented to be uncorrelated [9,10], it seems almost paradoxical that the centrality of an event could predict both. However, both outcomes are catalyzed by an event that is perceived as traumatic in a way that challenges existing cognitive paradigms (e.g., other people can be trusted, the world is safe, and good people cannot be harmed). Consistent with this supposition, judgments of greater event centrality have been associated with both a greater likelihood of intrusive ruminations about the event as well as increased probability that the person reports that the event violated one of their core beliefs [8]. However, less is known about whether judgments of event centrality, when applied to experiences of intimate partner infidelity (a non-Criterion A trauma that has been defined as a potentially traumatic event), will be associated with both of these post-trauma outcomes. This is the gap addressed in the current study.

## 1.2. Infidelity as a Potentially Traumatic Event

Infidelity within an intimate relationship is typically described as an interpersonal betrayal. However, its lasting effects can resemble those experienced by individuals with PTSD in response to other traumas [11,12]. Moreover, while generally understudied in comparison to traumatic events that are officially named by the *Diagnostic and Statistical Manual of Psychiatric Disorders* (*DSM*), infidelity and relational problems are often cited as the most distressing life events experienced by both LGBTQ+ and heterosexual populations [13]. The *DSM* defines a traumatic event to include "actual or threatened death, serious injury or sexual violence", and therefore, as per the *DSM* definition, infidelity cannot be characterized as a trauma. While the Substance Abuse and Mental Health Services Administration (SAMHSA) defines trauma as an "event, series of events, or set of circumstances that is experienced by an individual as physically or emotionally harmful or threatening and that has lasting adverse effects on the individual's functioning and physical, social, emotional, or spiritual well-being" [14] (p. 2). Likewise, in leading definitions of trauma-informed care, trauma is defined as constituting the three E's (event, experience, and effect). Therefore, this highlights the importance of considering an individual's experience of the event and not simply the event itself when determining whether something is traumatic [15].

That being said, in the previous literature, infidelity has been considered in a variety of ways, including as a nontraumatic stressful event, an interpersonal trauma, and a potentially traumatic event (PTE) [16,17]. The trauma model of infidelity [18] likens the subjective experience of infidelity to the experience of a trauma, a position that has been adopted by many other scholars in the field of relationship science [19–22]. Consistent with

this model, in the current work, we define infidelity as a potentially traumatic event (PTE) to denote the parallels infidelity can have with other experiences of trauma. Specifically, infidelity is typically unexpected and unwanted; it can violate core beliefs about the self, trusted others, important relationships, and the world; and it may even constitute sexual violence (due to heightened exposure to STIs without knowledge or consent), which can have negative physical health and mental health consequences. It is also an event that can evoke strong emotional responses, including fear, betrayal, shock, anger, sadness, and grief. That being said, there are dimensions of an infidelity experience that may make it look more or less like a traumatic event among those who experience it.

PTEs like infidelity have been shown to elicit similar levels of PTG as have other *DSM*-certified traumatic events [16]. Infidelity can function to rupture a strong interpersonal and romantic connection, which has been recognized as a vital component of an individual's psychological well-being [23]. In line with betrayal trauma theory, infidelity can also have long-standing and widespread impacts on an individual's ability to trust and receive comfort from others [24–27]. Betrayal is the cornerstone of experiences of infidelity, as it violates the trust that is crucial to adult romantic relationships. The intensity of the feeling of betrayal likely contributes to judgments of infidelity as a central experience in one's life.

Infidelity has also been linked with experiences of PTG, particularly as it influences the betrayed partner to redefine their values for what is sought in a romantic partner [28]. Experiences of infidelity within relationships have also been shown to lead betrayed partners to detach from the former relationship and become open to new romantic connections. This, in turn, can result in a better long-term interpersonal connection, a key component of PTG [28]. PTG as a result of infidelity has even been found in couples who remain together after infidelity, with forgiveness being a key predictor of that experience [29]. Some women who have been relationally betrayed have been shown to exhibit signs of PTG, with time being an important corollary to growth experiences [25]. Certain resources, like access to therapy, receiving psychoeducation, and experiencing support from and forgiveness for their partner, have been demonstrated to contribute to the development of PTG [25]. Overall, the experience of infidelity has been associated with both PTS and PTG, suggesting that designating this event as a PTE has validity.

### 1.3. Relationship Forms and Infidelity

While infidelity has been noted to contribute to PTS symptoms, this has primarily been demonstrated within marital relationships [25,27]. It is important to consider varying relationship dynamics, like lower levels of commitment, sexually permissive attitudes, and anxious attachment styles, that are present within college student couples that may put them at higher risk for experiencing infidelity [30,31], leaving them vulnerable to PTS. As demonstrated by Roos and colleagues [32], infidelity may produce PTS symptoms at a relatively high rate even in unmarried young adults, leading to poorer psychological health overall.

Relationship forms have emerged as a parsimonious way to consider the level of entanglement between romantic partners. Postulated to fall loosely along a continuum of commitment, these forms include casually dating, exclusively dating, and engaged/married. Casually dating has been defined as a couple engaging in a romantic connection that may include sexual contact without clear intentions or expectations for exclusivity or monogamy [33]. Exclusively dating can be defined by increased seriousness or commitment when couples have increased emotional and physical intimacy and exclusivity, often with the expectation of a future, long-lasting relationship [33]. Engaged or married relationships involve the most entanglement, including increased romantic and physical intimacy, often in conjunction with shared logistical considerations (i.e., living together, children, and joint finances). Furthermore, as couples make the transition from "casual" to "exclusive" and "exclusive" to "engaged or married", their intimacy and commitment increase [34]. These changes may be associated with differences in the perceived centrality of infidelity, as

betrayal trauma theory states that betrayal within a "closer" relationship is more impactful on the individual [24].

Importantly, however, the strength of the relationship between judgments of centrality and PTG and PTS may vary by relationship form. Theoretically, infidelity occurring in a dating relationship in young adulthood may be particularly impactful, as the individual experiencing infidelity is likely to have less experience in romantic relationships broadly [32]. Conversely, it could also be hypothesized that infidelity in more established and publicly solidified relationships, like marriage, may be perceived as more central and impactful, given that marital infidelity might necessitate a change in housing, finances, or child custody [35]. Dissolving a marriage post-infidelity may also evoke a greater sense of betrayal and/or generate greater community, religious, or social responses, and these, in turn, may generate altered cognitions about the self, partners, or the world. Thus, there is theoretical debate about whether and how perceptions of infidelity as a central experience within different relationship forms may differentially relate to PTG and PTS. The current study was designed to address this gap.

*1.4. The Current Paper*

Overall, the current paper seeks to extend the established associations between the centrality of an event and PTG and PTS outcomes in the context of infidelity, a PTE. Additionally, the current study adds to the research literature by evaluating the moderating effect of relationship form on the relationships between event centrality and PTG and PTS, respectively. This will be accomplished through the following research questions and related hypotheses:

To what extent is the experience of infidelity judged to be central? Does this vary by relationship form? Based on previous clinical literature on the nature of infidelity [18], judgements of centrality are expected. Judgements of centrality are also expected to vary by relationship form, such that infidelity will be judged as more central in engaged/married relationships compared to exclusively dating and casually dating relationships [36].

Does the perceived centrality of infidelity predict PTG within a sample of individuals who have recently experienced infidelity? Based on the literature on other traumas, a positive association is expected [5,8].

Does the perceived centrality of infidelity predict PTS within a sample of individuals who have recently experienced infidelity? Similarly, based on the literature on other traumas, a positive association is expected [1,5,8].

Does the relationship between the centrality of infidelity and PTG differ depending on the relationship form (casually dating, exclusively dating, or engaged/married) in which infidelity was experienced? A priori, relationship form was expected to moderate the positive relationship between the centrality of infidelity and PTG. Infidelity in longer-term relationships has been noted to provoke greater emotional responses and jealousy than infidelity in early-stage relationships [36]. Therefore, for forms that demonstrate more commitment (e.g., married/engaged), the relationship between the centrality of infidelity and PTG was expected to be weaker, and for forms that generally demonstrate less commitment (e.g., casually dating), the relationship between the centrality of infidelity and PTG was expected to be stronger.

Similarly, does the relationship between the centrality of infidelity and PTS differ depending on the form (casually dating, exclusively dating, or married/engaged) in which infidelity was experienced? Similarly, relationship form was expected to moderate the positive relationship between the centrality of infidelity and PTS. For forms that generally represent greater commitment (e.g., married/engaged), the relationship between the centrality of infidelity and PTS was expected to be stronger, and for forms that generally represent less commitment (e.g., casually dating), the relationship between the centrality of infidelity and PTS was expected to be weaker.

## 2. Methods

### 2.1. Participants and Procedure

All procedures were approved by the Institutional Review Board (IRB) at the University of North Carolina at Charlotte. Participants were recruited through two pathways. Primarily, students enrolled in introductory psychology courses at a large university in the Southeastern United States were invited to participate in this study as part of an undergraduate research subject pool. Secondarily, the survey was posted on social media platforms, including Facebook, and within other chat rooms that target individuals who have experienced infidelity. For all participants, the survey was administered online via Qualtrics. Participants completed the survey either on a personal computer or their phone. Participants were eligible to participate in this study if they had experienced infidelity within the last 12 months, were 18 or older at the time of consent, and were native English speakers. Inclusion criteria were not limited to a certain gender or sexual orientation, as infidelity experiences were expected to function similarly across these relationships [37,38].

An a priori power analysis, utilizing G*Power 3.1.9.7 software (G*Power, Düsseldorf, Germany) [39], was conducted to determine the necessary sample size to detect a moderation effect. This analysis indicated that to conduct a linear multiple regression examining the $R^2$ increase with two tested predictors and five overall predictors, a sample size of 132 would be necessary to detect a small effect and achieve 80% power.

A total of 247 participants agreed to participate; all self-reported having experienced relationship infidelity within the previous 12 months. The data were cleaned prior to analysis. Missing data within variables of interest were handled via mean imputation, such that means were calculated across participant responses for that individual scale. Listwise deletion was performed if participants were missing responses on more than 30% of that scale. As a result, (*n* = 70) participants were eliminated from the sample, leaving 177 participants. Participants were majority female (71%), and white (61%). Twenty percent of the sample identified as Black or African American, followed by Asian (10%), Other (6%), Native Hawaiian or Pacific Islander (1%), and American Indian/Alaskan Native (1%). A majority of participants identified as being non-Hispanic or Latino (85%). Participants' ages ranged from 18 to 67 years (*M* = 22.72, *SD* = 8.32). Participants primarily identified their sexual orientation as being straight or heterosexual (88%), bisexual (7%), gay or homosexual (3%), and other (1%). In addition, participants reported the classification of the relationship with the person who cheated on them as casually dating (19%), exclusively dating (71%), or engaged/married (10%).

### 2.2. Measures

**Centrality of the Event.** Centrality of the event was measured via the centrality of event scale (CES): short form [7], which is a 7-item measure designed to assess how central a major life crisis is to an individual's identity and life narrative. Items are measured on a scale from 1 (totally disagree) to 5 (totally agree). Examples of items from the CES include "I feel that this event has become part of my identity", and "This event has become a reference point for the way I understand myself and the world". The event in the current study was specified to be the experience of infidelity. A total score was calculated by averaging the items, with higher scores indicating a greater perceived centrality of infidelity compared to lower scores. The original authors reported excellent reliability for this measure in a sample of 707 undergraduate students (Cronbach's alpha = 0.94). Excellent reliability was also obtained in the current study (Cronbach's alpha = 0.91).

**Post-Traumatic Stress.** Post-traumatic stress was measured via the PTSD Checklist-5 [40], which is a 20-item measure used to assess the severity of trauma symptoms following a stressful event. The stressful event was specified as the reported infidelity. The scale is rated on a 1 (not at all) to 5 (extremely) scale. This scale assessed how much in the past month participants were bothered by experiences like "Feeling very upset when something reminded you of the stressful experience", and "Having difficulty concentrating?". A total symptom severity score was calculated by summing the scores for each of the 20 items, with

higher scores indicating greater symptom severity than lower scores. Excellent reliability was found for this measure in the current sample (Cronbach's alpha = 0.97).

**Post-Traumatic Growth**. Post-traumatic growth was measured via the Post-Traumatic Growth Inventory (PTGI) [41], which is a 21-item scale that measures the extent to which individuals report experiencing positive life changes in the aftermath of a traumatic experience, PTE, or life crisis. The current measure was adapted to specify that the potentially traumatic experience was infidelity. Items are rated on a six-point scale from 0 (I did not experience this change as a result of infidelity) to 5 (I experienced this change to a very great degree as a result of infidelity). Items were summed to create a single PTG score, with higher scores indicating greater levels of experiencing growth than lower scores. Excellent reliability was found for this measure in the current sample (Cronbach's alpha = 0.93).

**Relationship Form.** This question asked participants to classify the status of their previous relationship with the person with whom they experienced infidelity. Response options included "casually dating", "exclusively dating", "engaged", or "married". For the current study, the categories of "engaged" and "married" were combined due to sample size restrictions. The resulting three categories constituted levels of the moderator variable.

*2.3. Analytic Plan*

To examine research question one, descriptive analyses were conducted to determine mean ratings, standard deviations, and ranges of the centrality of infidelity. Additionally, a one-way ANOVA was conducted to compare mean ratings of the centrality of infidelity across relationship forms. To answer research questions two and three, bivariate regression analyses were utilized to determine the extent to which ratings of the centrality of infidelity, as the independent variable, were associated with PTG and PTS, as separate dependent variables. To address research questions four and five, moderation analyses were conducted utilizing PROCESS model 1 in SPSS for the described relationship, with relationship form (as defined by the following three categories: casually dating, exclusively dating, and married/engaged) tested as the moderator variable, with the centrality of infidelity as the predictor, and PTG and PTS as independent outcomes.

This study utilized a cross-sectional design. Prior to testing specific hypotheses, descriptive statistics (i.e., means, standard deviations, frequencies, and ranges) were conducted in IBM SPSS Statistics to ensure that expected relationships existed prior to regression and moderation analyses; see Table 1. Assumption tests related to regression and moderation were conducted, and assumptions of normality and linearity were met, as evidenced by P-P plots and normality histograms. No multicollinearity was observed between independent variables. Heteroscedasticity was observed within the current sample, but regression analyses are robust to violations of heteroscedasticity. To correct this, a heteroscedasticity-consistent standard error and covariance matrix estimator were used.

## 3. Results

As shown in Table 1, infidelity was indeed recognized as a central experience in participants' lives, indicated by means greater than three (range = 1–5) in 51.4% (*n* = 91) of the sample. Second, as predicted, centrality ratings varied significantly by relationship form, F (2, 174) = 4.80, *p* = 0.009. Correlations between variables of interest, centrality of infidelity, PTG, and PTS are shown in Table 2. In addition, associations between the centrality of infidelity and other theoretically relevant demographic variables of interest, including relationship length and age, were examined and are presented in Table 3.

**Table 1.** Descriptive statistics of major variables by relationship form.

| Variable | Casually Dating $n = 33$ (18.6%) $M$ ($SD$) | Exclusively Dating $n = 126$ (71.2%) $M$ ($SD$) | Engaged/Married $n = 18$ (10.2%) $M$ ($SD$) | F | $p$ |
|---|---|---|---|---|---|
| Centrality of Infidelity | 2.64 (1.15) | 3.00 (1.08) | 3.65 (1.32) | 4.80 | 0.009 ** |
| PTG | 91.22 (50.20) | 86.88 (37.31) | 89.97 (36.23) | 0.18 | 0.837 |
| PTS | 40.59 (17.57) | 45.63 (21.94) | 49.00 (25.96) | 1.05 | 0.353 |

** $p < 0.01$.

**Table 2.** Correlations among major study variables.

| Variable | 1 | 2 | 3 |
|---|---|---|---|
| 1. Centrality of Infidelity | - | 0.54 ** | 0.43 ** |
| 2. PTG | 0.54 ** | - | 0.40 ** |
| 3. PTS | 0.43 ** | 0.40 ** | - |

** $p < 0.01$.

**Table 3.** Correlations among relationship length, age, and centrality of infidelity.

| Variable | 1 | 2 | 3 |
|---|---|---|---|
| 1. Centrality of Infidelity | - | 0.09 | 0.20 ** |
| 2. Age | 0.09 | - | 0.30 ** |
| 3. Length of Relationship (with unfaithful partner) | 0.20 ** | 0.30 ** | - |

** $p < 0.01$.

### 3.1. Bivariate Regressions

To examine research question two, simple bivariate regression analyses were conducted with event centrality of infidelity as the predictor variable and PTG as the outcome variable. Results indicated that event centrality predicted a significant amount of variance in PTG, $F$ (1, 175) = 28.4, $p < 0.001$, $R^2 = 0.14$, $f^2 = 0.40$. Event centrality accounted for 13.5% of the variance in PTG. In support of the corresponding hypothesis, the slope coefficient for the centrality of infidelity predicting PTG was positive and statistically significant ($b = 0.37$, $p < 0.001$). Therefore, a one-unit increase in event centrality was associated with an increase of 0.37 in PTG.

To test research question three, a simple bivariate regression analysis was conducted with PTS as the outcome variable and event centrality as the predictor variable. Similarly, results indicated that event centrality predicted a significant amount of variance in PTS, $F$ (1, 175) = 40.02, $p < 0.001$, $R^2 = 0.19$, $f^2 = 0.48$. Event centrality accounted for 18.6% of the variance in PTS. In support of the corresponding third hypothesis, the slope coefficient for the centrality of infidelity in predicting PTS was positive and statistically significant ($b = 0.43$, $p < 0.001$). Therefore, a one-unit increase in event centrality was associated with an increase of 0.43 in PTS.

### 3.2. Moderation

To test research question four, the predictor variable (centrality of infidelity) was mean-centered. However, there was no need to center the moderator (relationship form) because it is categorical. Next, a multiplicative interaction term was created using the mean-centered predictor and the moderator variable. As the moderator is categorical, categories were dummy coded with casually dating as the reference group to examine differences in prediction across the three categories of the moderator variable. The variables and interaction terms were entered as predictors in a multiple regression with the outcome variable of PTG. The results indicated a statistically significant prediction of PTG using the

predictor variables combined, $F$ (5, 171) = 18.05, $p < 0.001$, $R^2 = 0.35$, $f^2 = 0.72$. It should be noted that analyses were also conducted to include relationship length as a covariate due to its theoretical relevance to relationship form, but its inclusion demonstrated no significant effect on model fit. Therefore, it was not included in the final analysis.

The centrality of infidelity was demonstrated to be predictive of PTG, $b = 30.09$, $t$ (171) = 6.02, $p < 0.001$. A significant difference in PTG between casually dating and exclusively dating groups was observed, $b = -15.35$, $t$ (171) = $-2.32$, $p = 0.02$. A nonsignificant difference in PTG between casually dating and engaged/married groups was observed, $b = -16.95$, $t$ (171) = $-1.62$, $p = 0.11$. Next, interactions were evaluated such that no interaction was observed between the comparison of casually dating versus exclusively dating by the centrality of infidelity, $b = -10.58$, $t$ (171) = $-1.86$, $p = 0.07$. However, a significant interaction was observed between the comparison of casually dating versus married/engaged by the centrality of infidelity, $b = -22.80$, $t$ (171) = $-2.92$, $p = 0.004$. The interaction term between the centrality of infidelity and relationship form significantly predicted PTG, $F$ (5, 171) = 4.29, $p = 0.015$, $\Delta R^2 = 0.033$. Simple slope analyses demonstrated differences in slopes for centrality to PTG at each level of relationship classification. For the casually dating relationship group, the slope for the centrality of infidelity predicting PTG was significant ($b = 30.09$, $p < 0.001$). Similarly, for the exclusively dating relationship group, the slope for the centrality of infidelity predicting PTG was significant ($b = 19.51$, $p < 0.001$). However, for those in the engaged/married group, the slope for the centrality of infidelity predicting PTG was nonsignificant ($b = 7.29$, $p = 0.23$). The regression lines for each classification were plotted in Figure 1 and demonstrate support for the fourth hypothesis. Specifically, the relationship between the centrality of infidelity and PTG appears to be stronger for those within the least serious relationship form, that is, casually dating.

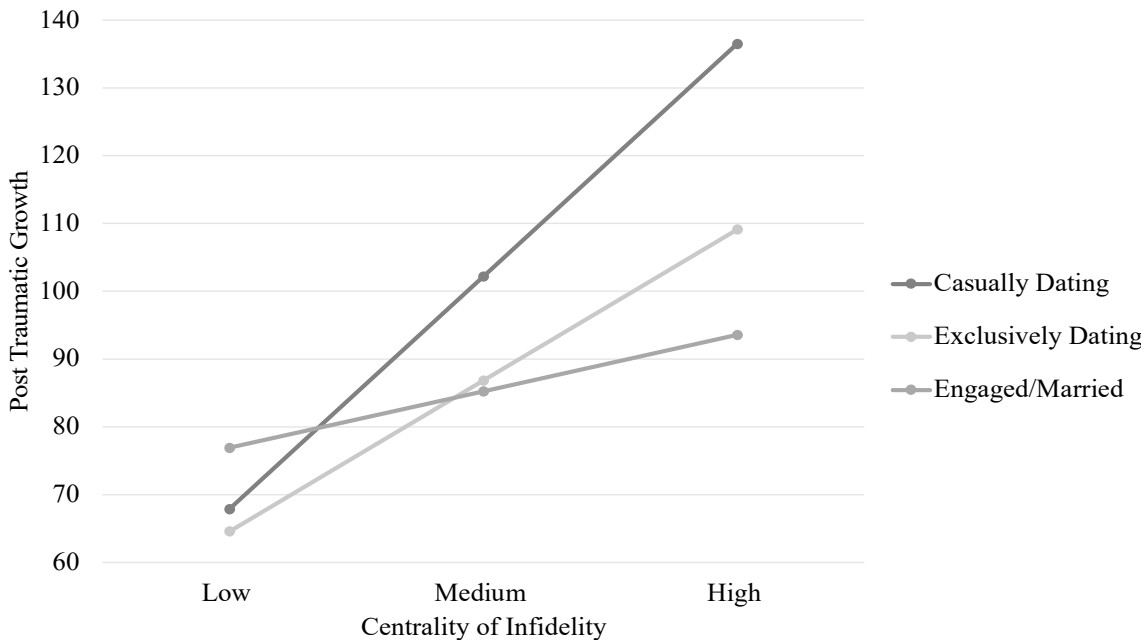

**Figure 1.** Simple slopes of the centrality of infidelity trauma predicting PTG for different relationship forms.

To test the fifth hypothesis, the same analyses were repeated with PTS as the dependent variable instead of PTG. The variables and interaction terms were entered as predictors in a multiple regression with the outcome variable of PTS. The results indicated a statistically significant prediction of PTS using the predictor variables combined, $F$ (5, 171) = 8.42, $p < 0.001$, $R^2 = 0.20$, $f^2 = 0.50$.

The centrality of infidelity was demonstrated to be predictive of PTS, $b = 9.83$, $t$ (171) = 3.25, $p = 0.001$. A nonsignificant difference in PTS between casually dating and exclusively

dating groups was observed, $b = 10.19$, $t$ (171) = 1.01, $p = 0.31$. Similarly, a nonsignificant difference in PTS between casually dating and engaged/married groups was observed, $b = -9.70$, $t$ (171) = $-0.59$, $p = 0.56$. Next, interactions were evaluated such that no interaction was observed between the comparison of casually dating versus exclusively dating by the centrality of infidelity, $b = -2.91$, $t$ (171) = $-0.85$, $p = 0.40$. Likewise, no interaction was observed between the comparison of casually dating versus married/engaged by the centrality of infidelity, $b = 2.23$, $t$ (171) = 0.47, $p = 0.64$. The interaction term between the centrality of infidelity and relationship form did not significantly predict PTS, $F$ (5, 171) = 1.02, $p = 0.36$, $\Delta R^2 = 0.01$, therefore indicating an insignificant moderating effect. Therefore, contrary to the fifth hypothesis, there was a nonsignificant moderation of relationship form on the centrality of infidelity predicting PTS.

## 4. Discussion

Infidelity is receiving greater recognition as a potentially traumatic interpersonal event within the clinical literature [11,32,42]. Even so, the outcomes of infidelity have been understudied. While clinical interests have often been focused on helping betrayed individuals recover from relationship dissolution as a result of infidelity [43], less is understood about how relationship characteristics and perceptions of infidelity influence an individual's own process of PTG or experience of PTS. The current study evaluated how the centrality of infidelity predicted experiences of PTG and PTS and how those associations varied by relationship form (i.e., casually dating, exclusively dating, and engaged/married). Infidelity has been understood as a relational betrayal that may violate core beliefs about trust in relationships [44]. Prior literature has demonstrated that events that violate core beliefs are more likely to be central to one's identity [1]. The experience of infidelity across the sample was recognized as central, and importantly, centrality was demonstrated to significantly differ by relationship form, with higher average centrality ratings occurring in the engaged/married group. These findings suggest that the legal and increasingly public nature of marriage or engagement may cause infidelity to be more central to those who experience it. Additionally, past research has demonstrated that prevalence rates of infidelity within less committed relationships are higher than in more committed relationships [45]. Perhaps, the more commonplace nature of infidelity within casually dating relationships leads this event to be judged as less central to one's life.

The second and third research questions considered the degree to which perceptions of the centrality of infidelity predict PTG and PTS. While the centrality of an event has been linked to experiences of PTG for traumas such as the death of a close friend or family member, serious accidents, physical or sexual assault, and wartime combat [8], this is the first known study to evaluate these relationships within the context of a relational PTE, specifically infidelity. As predicted, results demonstrated support for the first and second hypotheses that the centrality of infidelity is significantly related to both PTG and PTS, accounting for 13.5% and 18.6% of the variance in scores, respectively. Importantly, this study utilizes screening measures to understand self-reported symptoms and not to diagnose individuals based on these experiences. Even so, as the experience of the event is central to experiences of trauma [14], the results demonstrate clinically meaningful self-reported symptoms of PTS and PTG post-infidelity.

As noted in prior literature [8], the fact that centrality predicts both PTG and PTS may at first seem surprising. However, the same cognitive processes that occur in the face of infidelity can promote the thoughts and behaviors that contribute to both stress and growth [46]. Previous work has demonstrated the positive relationship between experiences of infidelity and PTG. Partner refinement, a component of PTG after infidelity, was associated with event centrality in previous work [28], demonstrating the role of centrality in engagement in growth behaviors. While some research has found that romantic relationship infidelity predicts experiences of PTG, this work is overall limited. The current study provides further evidence of this relationship, demonstrating the unique role of the centrality of infidelity in predicting the experiences of PTG. Heintzelman and

colleagues [29] examined a sample of individuals who remained in their relationships after infidelity, and found forgiveness within the relationship to be the only significant predictor of PTG. Taking the current results into account, centrality may also play a vital role in the progression to PTG.

PTS symptoms following relationship infidelity have also been demonstrated in previous literature. Roos and colleagues [32] found that unmarried young adults experienced PTSD post-infidelity at a relatively high rate, and those symptoms were associated with overall decreased psychological health. Similarly, a sample of combat veterans who experienced infidelity while deployed were shown to experience greater PTSD symptomatology post-deployment than their counterparts who did not experience infidelity [47]. In practice, clinical psychologists who specialize in couples' treatment have long seen the potentially traumatic effects of infidelity on relationship health and have developed effective treatment methodologies [48]. Results from the current work demonstrate that perceptions of centrality may play a role in an individual's experience of PTS post-infidelity. Drawing from PTG theory, it is possible that there are corresponding processes of intrusive then deliberative rumination that mirror PTS and PTG outcomes. Being able to deliberately ruminate on their experience of infidelity could help the betrayed partner to detach from their previous relationship, redefine what they desire in a romantic partner, and become open to new and improved romantic connections [28].

Secondarily, the current study aimed to understand the moderating effect of relationship form on the relationships between the centrality of infidelity, PTG, and PTS. Three relationship forms were considered: casually dating, exclusively dating, and married/engaged. These relationship forms were expected to vary internally based upon the degree of seriousness of commitment and entanglement and vary externally in terms of expectations for monogamy and the public nature of the relationship. Relationship form demonstrated a significant moderating effect in the relationship between the centrality of infidelity and PTG, such that the strongest relationship between these two constructs was found among those reporting that infidelity occurred in casual dating relationships. This result demonstrates that judging early dating infidelity as central, might set one up for growth in romantic relationships, perhaps through future partner refinement and greater use of effective communication [28].

However, relationship form did not significantly moderate the relationship between the centrality of infidelity and PTS. Therefore, only hypothesis four was supported. This provides important evidence that relationship form may be more meaningful in predicting PTG compared to PTS. Replication of this result will be necessary. Additionally, this begs the question: what is it about the centrality of infidelity in casual dating relationships that more strongly predicts post-traumatic growth? It could be that experiencing infidelity in a casually dating relationship may inherently offer more opportunity for PTG, such that these kinds of relationships often occur earlier in life, and often do not face situational repercussions like division of finances or responsibility for children. Less of those repercussions may allow more opportunity to learn from the experience of infidelity about what is desired within a future partner, and perhaps even greater ability to recognize signs of infidelity.

Much of the existing literature examining infidelity and its consequences has focused on marital couples [25], failing to consider the large portion of individuals who experience infidelity in nonmarital dating relationships. Typically, married individuals present more readily for couples' treatment, which has made them more accessible to clinical relationship research [32]. Although these populations may be more readily available, marital research fails to capture all relationships in which infidelity is experienced. As recent trends indicate that individuals are more often delaying marriage by an average of seven years, long-term dating relationships and cohabitation are on the rise [49,50]. Due to the shift in the demographics of relationships across the U.S., nonmarital dating relationships are of increasing clinical interest [51], and it is important to be able to capture the experiences of those populations within research.

The results of this study have several possible clinical implications. Firstly, while infidelity is not considered a trauma by *DSM* definition, these results demonstrate that infidelity does generate symptoms of PTS and PTG, generating two important questions: (1) are the *DSM*-defined criteria for a traumatic event too strict? and (2), as clinicians, how do we treat effects related to potentially traumatic events, like infidelity? Broadening the *DSM* criteria for what qualifies as trauma (Criterion A) has both clinical advantages and disadvantages that are important to note. For example, loosening the trauma criteria may allow more PTE sufferers to have access to treatment; it may also normalize the long-term symptoms some betrayed partners experience. Greater investigation into useful clinical treatments for PTS symptoms that are associated with common PTEs may also facilitate a more inclusive definition of trauma. For example, further investigation is warranted to determine if gold-standard trauma treatments (e.g., cognitive processing therapy, or CPT) are appropriate and useful for this type of symptom presentation post-infidelity. Conversely, loosening the *DSM* criteria for what constitutes trauma may lead to clinical overdiagnosis, reduced cultural expectations of adapting and adjusting to stressors, and unwarranted under-responsiveness to those recovering from a life-threatening trauma. Consideration of the specificity and scope of diagnostic criteria in the context of their clinical utility is an ongoing and important conversation.

However, as infidelity is one of the most common and most difficult problems to treat within couples therapy [38], understanding the centrality of infidelity may be an avenue through which clinicians can address symptom severity within the relationship. Recognizing one's perceived centrality of infidelity may be significant for some clients as a way to understand and make meaning of this betrayal trauma, providing a pathway to growth, as meaning-making has been understood to lead to higher levels of PTG [8]. As we noted that PTG and PTS often co-occur, it is also important for clinicians to attend to the valence of the centrality of the experience of infidelity, as centrality can be perceived positively or negatively [1,8]. Our examination of relationship form also gives clinical evidence that these pathways to PTG and PTS occur across all forms of romantic entanglement, which may be helpful for those who work with populations with high levels of nonmarital relationships (e.g., college counseling center mental health professionals). Finally, moderation results demonstrate that it may be more difficult for individuals in more committed relationships to achieve PTG. Attending to the centrality of infidelity may only be a piece of the puzzle that works towards clinical improvement.

Limitations to this study should be noted. Primarily, while there was a focus on different relationship forms, there is still limited variability in the forms of relationships examined (e.g., casually dating, exclusively dating, and married/engaged). As Jamison and Sanner [33] demonstrated in their recent study using relationship history interviews of 35 young adults, there are many nuanced relationship forms, including time-bound dating, hookups, or romantic experimentation, that may differentially impact one's experience of infidelity and progression towards outcomes like PTG or PTS. Greater consideration of these evolving and changing relationship dynamics should be made in future research. Additionally, sample sizes were relatively small. A greater variety of relationship forms is likely to be captured by a larger sample size. In particular, small sample sizes within groups of the moderator variable reduce the overall power of the moderation analyses by increasing the likelihood of type 1 error, but this is less of a concern in the current sample due to the large observed effect sizes. In general, results should be interpreted with the impact of the sample size in mind. Another potential limitation of this study is the use of a cross-sectional design to assess the experiences of PTG and PTS. A longitudinal design would facilitate consideration of whether and how PTG or PTS symptoms develop over time after the experience of infidelity. While the centrality of infidelity appears to be a useful predictor of PTG and PTS symptoms, the Centrality of Event Scale utilized in this study does not assess the way in which infidelity was understood (i.e., positively or negatively), and incorporation of that measurement within future studies would provide

information as to how valanced centrality ratings may differentially affect experiences of PTG or PTS.

Nonetheless, this study provides empirical support for defining infidelity as a potentially traumatic event that can generate PTS and PTG in some individuals who were betrayed. Furthermore, relationship form and judgments of centrality are both important considerations as these processes unfold over time.

**Author Contributions:** Conceptualization, B.N.J., V.L.O. and J.L.-R.; methodology, V.L.O. and J.L.-R.; formal analysis, B.N.J.; investigation, V.L.O. and J.L.-R.; resources, B.N.J. and V.L.O.; data curation, V.L.O. and J.L.-R.; writing—original draft preparation, B.N.J.; writing—review and editing, B.N.J., J.L.-R. and V.L.O.; supervision, J.L.-R.; project administration, V.L.O. All authors have read and agreed to the published version of the manuscript.

**Funding:** This research received no external funding.

**Institutional Review Board Statement:** The study was approved by the Institutional Review Board of the University of North Carolina Charlotte (protocol code: IRBIS-19-0706 and date of approval: 8 April 2020).

**Informed Consent Statement:** Informed consent was obtained from all subjects involved in the study, and Institutional Review Board approval was obtained prior to conducting this study.

**Data Availability Statement:** The data presented in this study are available on request from the corresponding author. The data are not publicly available due to the confidentiality of participant information.

**Acknowledgments:** This work was supported by the Salisbury VA Health Care System, Mid-Atlantic (VISN 6) Mental Illness Research, Education, and Clinical Center (MIRECC), and the Department of Veterans Affairs Office of Academic Affiliations Advanced Program in Mental Illness, Research, and Treatment. This work was also supported by a summer research graduate assistantship awarded by the Health Psychology Doctoral Program at the University of North Carolina, Charlotte.

**Conflicts of Interest:** The authors declare no conflict of interest.

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
