# Peer review of "Judgments of Event Centrality as Predictors of Post-Traumatic Growth and Post-Traumatic Stress after Infidelity: The Moderating Effect of Relationship Form"

_traumacare, doi:10.3390/traumacare3040021_

Round 1

Reviewer 1 Report

The submitted manuscript addresses an important type of interpersonal traua, namely infidelity. The Authors focused on the associations between centrality of infidelity, form of relationships in which an individual experienced an infideity and post-traumatic growth and stress. The goals of the study were justified by the introduction. Although, the goals and theory are strengths of the manuscript, I have doubts regarding some methodological issues.

#1. The tables summarizing the associations between studied variables (e.g., correlations between PTS and PTG) is missing.

#2. The Authors should present details of their sample size calculations. 18 participants in engaged/married relationships seem to be a low number to detect even moderate effects of the associations between centrality and PTS or PTG. Moreover, the Authors should also refer to stability of correlation estimation which requires smaples of 250 participants or more (Schonbrodt ¨ & Perugini, 2013) and power to detect effects in ANOVA.

#3. Effect sizes and post-hoc tests are missing in ANOVA (eta squares). 

#4. Did the Authors control for relationship length? At least for descriptive purposes, the Authors can present associations between participants' age, and their relationships' lenght.

#5. Some issues in discussion mix the results with hypotheses about its appearance. It will be better to directly state which are the Auhtors hypotheses about possible mechanisms of the obtained results.

#6. Regarding the analysis, if the PTS and PTG are associated, it will be worth to consider SEM (e.g., multigroup SEM). However, the sample size of subgroups is a barrier in this type of an analysis.

Author Response

Response to Reviewer 1 Comments

1. Summary

Thank you very much for taking the time to review this manuscript. Please find the detailed responses below and the corresponding revisions/corrections made throughout the manuscript.

2. Questions for General Evaluation

Reviewer’s Evaluation

Response and Revisions

Does the introduction provide sufficient background and include all relevant references?

Yes

We appreciate the reviewer’s belief in the strength of our goals and theoretical justification for this work. We hope that amendments to recognize the limitations of our methodological design and conclusions have addressed the reviewer’s concerns.

Are all the cited references relevant to the research?

Yes

Is the research design appropriate?

Can be improved

Are the methods adequately described?

Yes

Are the results clearly presented?

Yes

Are the conclusions supported by the results?

Can be improved

3. Point-by-point response to Comments and Suggestions for Authors

The submitted manuscript addresses an important type of interpersonal traua, namely infidelity. The Authors focused on the associations between centrality of infidelity, form of relationships in which an individual experienced an infideity and post-traumatic growth and stress. The goals of the study were justified by the introduction. Although, the goals and theory are strengths of the manuscript, I have doubts regarding some methodological issues.

Comments 1: The tables summarizing the associations between studied variables (e.g., correlations between PTS and PTG) is missing.

Response 1: We thank the reviewer for pointing this out, we agree this is important descriptive information and therefore, we have added Table 2 to summarize these associations.

Comments 2: The Authors should present details of their sample size calculations. 18 participants in engaged/married relationships seem to be a low number to detect even moderate effects of the associations between centrality and PTS or PTG. Moreover, the Authors should also refer to stability of correlation estimation which requires samples of 250 participants or more (Schonbrodt ¨ & Perugini, 2013) and power to detect effects in ANOVA.

Response 2: We thank the reviewer for this comment. They aptly point out that power of the association varies by size of group. In our a-priori power analysis which considers all predictors (including the interaction at group levels), our sample was demonstrated to be sufficiently powered. Even so, we recognize the limitation a small group provides and therefore have added that to our limitations section in the discussion. As effect sizes for the married/engaged group were demonstrated to be large, we believe this concern was mitigated.

Comments 3: Effect sizes and post-hoc tests are missing in ANOVA (eta squares). 

Response 3: We thank the reviewer for pointing this out. Effect sizes and appropriate post-hoc tests have been added throughout the results section.

Comments 4: Did the Authors control for relationship length? At least for descriptive purposes, the Authors can present associations between participants' age, and their relationships' length.

Response 4: Thank you for this comment. We did not control for relationship length in original analyses but have re-run our analyses to include relationship length as a covariate and this had no impact on study results. We have included a sentence in our results to demonstrate this. Additionally, we agree that the association between a relationship length, age, and Centrality of Infidelity is theoretically relevant and therefore have included table 3 to illustrate those associations.

Comment 5: Some issues in discussion mix the results with hypotheses about its appearance. It will be better to directly state which are the Auhtors hypotheses about possible mechanisms of the obtained results.

Response 5: We thank the reviewer for pointing this out. We have amended the discussion to provide greater clarity on our hypotheses about the mechanisms by which our results are functioning.

Comment 6. Regarding the analysis, if the PTS and PTG are associated, it will be worth to consider SEM (e.g., multigroup SEM). However, the sample size of subgroups is a barrier in this type of an analysis.

Response 6: We thank the reviewer for this insight. We agree that this would be something important to consider if our 3-group sample size could support it. Unfortunately, our current data would not allow for this type of analysis, nonetheless, we appreciate this consideration.

4. Response to Comments on the Quality of English Language

Point 1:

None

5. Additional clarifications

None

Reviewer 2 Report

I thank the authors for their study. While I understand their interest in infidelity as a trauma, I feel the field of trauma already has a solution for that: adjustment disorder. If infidelity is non-traumatic stressor, it cannot be studied using PTS-measures, but through adjustment disorder-scales. The argument that infidelity can "feel like a trauma" might be true, but then you are discussing the "perception of trauma", or in other words, the colloquial use of trauma. Not the scientific use. There is no indication at all in criterion A of the DSM-5 that it might be a trauma. You actually had a better chance of making your case using ICD-11 definitions of trauma. 

Also, measuring PTSD to prove a trauma is a reverse logic. So, it is not because you find PTSD-rates that it is correct - because the problem remains that infidelity is not recognized as a trauma.

Furthermore, the authors make the argument that criterion A has changed over time. Indeed, it has actually become more narrow again in DSM-5. DSM-IV definition supported the statement of the authors somewhat, which had very broad inclusiosn. But because they weren't that useful clinically speaking, they were made more narrow again. There is no life-threatening exposure in infidelity. There is violence. No sexual assault. It is almost to the point of insulting to people who experience sexual assault and disasters that this would be considered a trauma. Therefore, I have to advise to reject this paper. 

"Two vital outcomes of trauma include experiences of post-traumatic growth (PTG) 32 and symptoms of posttraumatic stress"

This is misleading, as most trauma won't lead to anything. 70% of the world population experiences trauma at least once in their life. 

"n contrast, PTS is characterized by intrusive and unwanted memories of the 35 traumatic experience, heightened depression and anxiety, uncomfortable physical reac- 36 tions including sweating and heart racing in response to trauma reminders or triggers, 37 and more"

Heightened depression and anxiety? Diagnostically speaking untrue.

"Infidelity within an intimate relationship is conceptualized as an interpersonal 73 trauma, as its lasting effects resemble those experienced by individuals with PTSD in re- 74 sponse to other traumas" 

This is simply untrue. If "infidelity" is a trauma, then what do the authors see as "non-traumatic stressors"? What does then fit adjustment disorder? 

" Moreover, while generally understudied in comparison 75 to traumatic events that are officially named by the Diagnostic and Statistical Manual of 76 Psychiatric Disorders (DSM), infidelity, and relational problems, are often cited as the most 77 distressing life events experienced by both LGBTQ+ and heterosexual populations"

This is different from a trauma. Distress can be caused by stressors. Not all stressors are therefore trauma.

"In 78 addition, nontraumatic stressful events, like infidelity, have been shown to elicit similar 79 levels of PTG compared to traumatic events"

I don't understand. Do you view it as a traumatic event or not? As you state here, it is a non-traumatic stressor. Therefore, PTSD cannot be diagnosed, by definition. 

"Importantly, the DSM under- 82 standing of criterion A trauma, and PTSD broadly has evolved over time (e.g., PTSD is no 83 longer understood as an anxiety disorder as it was in DSM-IV). Therefore, it is essential to 84 consider conceptualizations of trauma and events that lead to post-trauma outcomes as 85 evolving, such that relational discord, like infidelity, may indeed constitute trauma."

Again: adjustment disorder. I would strongly advise the authors to look up adjustment disorders in both the ICD-11 and the DSM-5-TR. 

So, in general, I completely disagree with the conceptualization of this study. You cannot say "It's not a trauma, but let's pretend it's a trauma anyway". 

The results also are very odd for the PCL-5. It goes from 0 to 80, with a cutoff generally at 31 or 33. That means that on average respondents had PTSD... In general, the results are also quite unclear. 

(I also have an issue with how the methods are structuralized - already mentioning how participants you have is odd, because you have not yet described your methods at that point). 

Author Response

Response to Reviewer 2 Comments

1. Summary

Thank you very much for taking the time to review this manuscript. Please find our response below and the corresponding revisions/corrections within the resubmitted files.

2. Questions for General Evaluation

Reviewer’s Evaluation

Does the introduction provide sufficient background and include all relevant references?

Can be improved

Are all the cited references relevant to the research?

Must be improved

Is the research design appropriate?

Yes

Are the methods adequately described?

Must be improved

Are the results clearly presented?

Can be improved

Are the conclusions supported by the results?

Must be improved

3. Response to Comments and Suggestions for Authors

Comments 1:

I thank the authors for their study. While I understand their interest in infidelity as a trauma, I feel the field of trauma already has a solution for that: adjustment disorder. If infidelity is non-traumatic stressor, it cannot be studied using PTS-measures, but through adjustment disorder-scales. The argument that infidelity can "feel like a trauma" might be true, but then you are discussing the "perception of trauma", or in other words, the colloquial use of trauma. Not the scientific use. There is no indication at all in criterion A of the DSM-5 that it might be a trauma. You actually had a better chance of making your case using ICD-11 definitions of trauma. 

Also, measuring PTSD to prove a trauma is a reverse logic. So, it is not because you find PTSD-rates that it is correct - because the problem remains that infidelity is not recognized as a trauma.

Furthermore, the authors make the argument that criterion A has changed over time. Indeed, it has actually become more narrow again in DSM-5. DSM-IV definition supported the statement of the authors somewhat, which had very broad inclusion. But because they weren't that useful clinically speaking, they were made more narrow again. There is no life-threatening exposure in infidelity. There is violence. No sexual assault. It is almost to the point of insulting to people who experience sexual assault and disasters that this would be considered a trauma. Therefore, I have to advise to reject this paper. 

"Two vital outcomes of trauma include experiences of post-traumatic growth (PTG) 32 and symptoms of posttraumatic stress"

This is misleading, as most trauma won't lead to anything. 70% of the world population experiences trauma at least once in their life. 

"In contrast, PTS is characterized by intrusive and unwanted memories of the traumatic experience, heightened depression and anxiety, uncomfortable physical reactions including sweating and heart racing in response to trauma reminders or triggers, and more"

Heightened depression and anxiety? Diagnostically speaking untrue.

"Infidelity within an intimate relationship is conceptualized as an interpersonal trauma, as its lasting effects resemble those experienced by individuals with PTSD in response to other traumas" 

This is simply untrue. If "infidelity" is a trauma, then what do the authors see as "non-traumatic stressors"? What does then fit adjustment disorder? 

" Moreover, while generally understudied in comparison to traumatic events that are officially named by the Diagnostic and Statistical Manual of Psychiatric Disorders (DSM), infidelity, and relational problems, are often cited as the most distressing life events experienced by both LGBTQ+ and heterosexual populations"

This is different from a trauma. Distress can be caused by stressors. Not all stressors are therefore trauma.

"In addition, nontraumatic stressful events, like infidelity, have been shown to elicit similar levels of PTG compared to traumatic events"

I don't understand. Do you view it as a traumatic event or not? As you state here, it is a non-traumatic stressor. Therefore, PTSD cannot be diagnosed, by definition. 

"Importantly, the DSM understanding of criterion A trauma, and PTSD broadly has evolved over time (e.g., PTSD is no 83 longer understood as an anxiety disorder as it was in DSM-IV). Therefore, it is essential to onsider conceptualizations of trauma and events that lead to post-trauma outcomes as evolving, such that relational discord, like infidelity, may indeed constitute trauma."

Again: adjustment disorder. I would strongly advise the authors to look up adjustment disorders in both the ICD-11 and the DSM-5-TR. 

So, in general, I completely disagree with the conceptualization of this study. You cannot say "It's not a trauma, but let's pretend it's a trauma anyway". 

The results also are very odd for the PCL-5. It goes from 0 to 80, with a cutoff generally at 31 or 33. That means that on average respondents had PTSD... In general, the results are also quite unclear. 

(I also have an issue with how the methods are structuralized - already mentioning how participants you have is odd, because you have not yet described your methods at that point).

Response 1:

We thank the reviewer for this reaction to our paper and we are dismayed that our writing was potentially distressing or insulting to those who have experienced DSM defined experiences of trauma. We recognize this may be a reaction that researchers who study trauma and those who experience such events may have, and in response we have done the following things:

1.       We have re-written our paper to consider infidelity a potentially traumatic event (PTE) instead of a trauma due to criterion similarities including that infidelity is typically unexpected and unwanted; it can violate core beliefs about the self, trusted others, important relationships, and the world; and it may even constitute sexual violence (due to heightened exposure to STIs without knowledge or consent) which can have health and mental health consequences. It is also an event that can evoke strong emotional responses to include fear, betrayal, shock, anger, sadness, and grief. That being said, there are dimensions of an infidelity experience that may make it look more or less like a traumatic event among those who experience it. 

2.       While we appreciate the reviewer’s consideration of infidelity as adjustment disorder, we disagree that this fits well in this category as it may have lasting effects long outside the 3-month designated window as defined by the DSM due to disruption of core beliefs triggered by experiences of infidelity. We also agree that infidelity does not fit under the current DSM definition of trauma but believe that adjustment disorder does not accurately portray the clinical impacts of infidelity as it would define the symptoms to be “out of proportion with the intensity of the stressor”, which we do not feel is accurate. We hope our consideration of infidelity as a PTE clarifies our position and is acceptable to the reviewer.

3.       We have included appropriate citations to denote the ways in which infidelity has been considered within previous literature and called upon the Trauma Model of Infidelity as a major theoretical influence on this work.

4.       We have clarified our writing to denote PTG and PTS are two possible, but not common results, of trauma/PTE. We agree with the reviewer that a relatively small sample of individuals who experience a PTE experience these outcomes.

5.       We have added in material in the discussion to denote our belief in the clinical utility of considering Infidelity a PTE (p. 11 line 462 - 494).

6.       The methods have been restructured to respond to the reviewer’s comment about method structure. We appreciate this comment and believe the new structure better represents the order of operations.

Round 2

Reviewer 1 Report

The Authors modifications resulted in improvement of the manuscript which could be now suggested for publication.

Author Response

We thank this reviewer for their considerations and contribution to improving this manuscript. We believe as a result of these changes, we have an enhanced version of our paper. 

Reviewer 2 Report

Thank you for the revision. However, as I previously mentioned, I believe there are simply too many assumptions being made, both on what screening can prove as well as diagnostical issues. If it is "non-traumatic stressor", then by definition it is not traumatic. You might argue that it is traumatic, but where does trauma and non-traumatic stressor begin? The authors for example state that infidelity can include "sexual violence". If that's the case, of course it is a trauma. Why? Because sexual violence is forced upon the integrity of the person. That something challenges your core believes is simply not part of the trauma definition of the DSM-5. So, post-traumatic growth in this case is maybe just... moving on. I always feel it is almost insulting to those who have been sexually assaulted, who survived human trafficking et cetera, to put those experiences on the same level as infidelity, which has many different dimensions where you cannot really say a priori that it is shattering of core believes, or of anything really. And I understand that other authors might have used a broadening of what trauma is to include infidelity, but that says more of the generallly poor peer review, and poor knowledge of other researchers of trauma. You should never really on articles to define the core of your study if it relates to such psychiatric concepts. Because researchers will always broaden terms to fit their own goals. That does not mean it is right, or useful. I can find studies that show that being bored is traumatic. Or working too much. The concept becomes useless really fast if we really on how researchers conceptualize it, when for the purpose of screening studies, criterion A (or the inclusion for ICD-11 PTSD) are very useful to allow comparibility between studies. However, if every researchers starts to make their own version of trauma, then, well, nobody is researching anything that is useful in terms of bettering the treatment of victims of trauma. So, again, my apologies to advise rejection, but from my own viewpoint I cannot in good consciousness recommend acceptance. For future publications, I would personally recommend that you avoid the argument that infidelity is a trauma. You can always use PTG, but explain perhaps that it is used as a way of seeing how people recover from infidelity - because, infidelity is of course impactful on someone's life. 

Author Response

We appreciate that the reviewer has taken the stance that infidelity is not a trauma. We also understand their concern that widening the parameters of what constitutes a trauma potentially leads to both a lack of comparability among studies of trauma and may also be seen as “insulting” to those who consider their trauma to be more severe than infidelity (i.e., victims of sexual assault, human trafficking). We do not expect to change this reviewer’s mind on this topic.

We are offering the following as our rebuttal while we simultaneously wish to point out the numerous ways we have already addressed this reviewer’s concern in the current paper. First, we remind this reviewer that having a particular qualifying trauma (Criterion A) has not been proven to be definitive in the study of PTSD, leading to much debate over the years as to what should and shouldn’t be included in Criterion A (e.g., see Weathers & Keane, 2007, The Criterion A problem revisited: controversies and challenges in defining and measuring psychological trauma DOI: 10.1002/jts.20210 as just one example. In this article, they “provide an update on the Criterion A problem, with particular emphasis on the evolution of the DSM definition of the stressor criterion and the ongoing debate regarding broad versus narrow conceptualizations of traumatic events.”) Second, as we clearly note in our current paper, there are ways infidelity might manifest that would clearly constitute a Criterion A trauma even as it is narrowly delineated (particularly if the infidelity involves the transmission of a STI, received surprisingly with an unwanted violation of monogamy). Third, while the reviewer is focused on the DSM definition of trauma, we note that many leading funding agencies utilize different definitions of trauma. For example, SAMHSA defines trauma as an “event, series of events, or set of circumstances that is experienced by an individual as physically or emotionally harmful or threatening and that has lasting adverse effects on the individual's functioning and physical, social, emotional, or spiritual well-being”. In our opinion, which is shared by leading scholars in the field of couple and family relationships and is noted in our paper, infidelity clearly qualifies as a trauma utilizing this definition. Likewise, in leading definitions of trauma-informed care, trauma is defined as constituting the three E’s (event, experience, effects). Consider SAMHSA’s Tip 57, which states, “It is not just the event itself that determines whether something is traumatic, but also the individual’s experience of the event. Two people may be exposed to the same event or series of events but experience and interpret these events in vastly different ways. Various biopsychosocial and cultural factors influence an individual’s immediate response and long-term reactions to trauma. For most, regardless of the severity of the trauma, the immediate or enduring effects of trauma are met with resilience—the ability to rise above the circumstances or to meet the challenges with fortitude.” Thus, we argue that this reviewer’s over focus on the event (infidelity) and minimization of the interpretation/experience (which certainly can include altering the integrity of the person) and its effects (which are measured in the current study as substantial) is also upsetting and inappropriately minimizing to some who have experienced infidelity as traumatizing.

Nonetheless, we believe we have revised our current paper in a number of ways to address this reviewer’s concerns to the extent possible. Namely, we have refrained from labelling infidelity as a trauma. Instead, we consistently refer to it as a potentially traumatic event or PTE. Thus, we use traumatic as a adjective rather than a noun, in deference to Criterion A. Next, we include the noted relationship scholars who have also advanced the argument that infidelity can be experienced as traumatic. Third, we report the effects of experiencing this event, which is consistent with the position taken by SAMHSA. Fourth, we believe that understanding the impact of infidelity, which in many cases is substantial, is important clinically. We point out the implications of our findings for clinical practice with couples. Fifth, we highlight how betrayal trauma theory, articulated by Jennifer Freyd, postulates that potentially traumatic events that involve betrayal by our trusted others, is particularly harmful as it often violates our sense of self, others, and the world. This violation is known to be one of the cognitive factors that leads to symptoms of PTSD. Sixth, we remind the reviewer that we are not diagnosing individuals with PTSD, we are measuring self-reported symptoms. These activities are not synonymous. Finally, we have included a paragraph in our discussion section which brings the reviewer’s concerns to the readers’ attention in order to facilitate further discussion/transparency about this issue. We do not know of any other way to address this reviewer’s concerns.